# Diagnostic Value of IgM and IgG Detection in COVID-19 Diagnosis by the Mobile Laboratory B-LiFE: A Massive Testing Strategy in the Piedmont Region

**DOI:** 10.3390/ijerph18073372

**Published:** 2021-03-24

**Authors:** Omar Nyabi, Mostafa Bentahir, Jérôme Ambroise, Bertrand Bearzatto, Nawfal Chibani, Benjamin Smits, Jean François Durant, Aleksandr Vybornov, Olivier Thellin, Benaissa El Moualij, Jean-Luc Gala

**Affiliations:** 1Center for Applied Molecular Technologies (CTMA), Institut de Recherche Expérimentale et Clinique (IREC), Université Catholique de Louvain, 1348 Woluwe saint-Lambert, Belgium; mostafa.bentahir@uclouvain.be (M.B.); jerome.ambroise@uclouvain.be (J.A.); bertrand.bearzatto@uclouvain.be (B.B.); nawfal.chibani@uclouvain.be (N.C.); smith.benjamin@uclouvain.be (B.S.); jean-francois.durant@uclouvain.be (J.F.D.); Aleksandr.vybornov@uclouvain.be (A.V.); jean-luc.gala@uclouvain.be (J.-L.G.); 2Centre de Recherche sur les Protéines Prions (CRPP) ULiège, Quartier Hôpital, 15, Avenue Hippocrate, B 4000 Liège, Belgium; o.thellin@uliege.be (O.T.); b.elmoualij@uliege.be (B.E.M.)

**Keywords:** SARS-CoV-2, Public Health Preparedness, emerging biological threats

## Abstract

Coronavirus disease 2019 (COVID-19) is an acute infectious disease caused by the novel coronavirus (SARS-CoV-2) identified in 2019. The COVID-19 outbreak continues to have devastating consequences for human lives and the global economy. The B-LiFe mobile laboratory in Piedmont, Italy, was deployed for the surveillance of COVID-19 cases by large-scale testing of first responders. The objective was to assess the seroconversion among the regional civil protection (CP), police, health care professionals, and volunteers. The secondary objective was to detect asymptomatic individuals within this cohort in the light of age, sex, and residence. In this paper, we report the results of serological testing performed by the B-LiFe mobile laboratory deployed from 10 June to 23 July 2020. The tests included whole blood finger-prick and serum sampling for detection of SARS-CoV-2 spike receptor-binding domain (S-RBD) antibodies. The prevalence of SARS-CoV-2 antibodies was approximately 5% (294/6013). The results of the finger-prick tests and serum sample analyses showed moderate agreement (kappa = 0.77). Furthermore, the detection rates of serum antibodies to the SARS-CoV-2 nucleocapsid protein (NP) and S-RBD among the seroconverted individuals were positively correlated (kappa = 0.60), at least at the IgG level. Seroprevalence studies based on serological testing for the S-RBD protein or SARS-CoV-2 NP antibodies are not sufficient for diagnosis but might help in screening the population to be vaccinated and in determining the duration of seroconversion.

## 1. Introduction

The severe acute respiratory syndrome coronavirus (SARS-CoV) and the Middle East respiratory syndrome coronavirus (MERS-CoV) were first isolated in China and the Middle East, respectively. The transmission of these viruses from animals to humans has led to severe respiratory diseases in humans, namely SARS and MERS, in endemic areas [1]. In December 2019, a new type of coronavirus was discovered in Wuhan, Hubei Province, China. This virus appears to be highly infectious with human-to-human transmission [2,3]. The disease, now termed coronavirus disease 2019 (COVID-19), spread rapidly worldwide, resulting in a pandemic. The major phenotype of COVID-19 is the severe acute respiratory distress syndrome (ARDS), similar to that observed in cases of SARS and MERS [4,5]. If steroids are able to decrease the related systemic inflammation in severe cases, convalescent plasma with high concentration of anti-SARS-CoV-2 IgG levels was shown to be associated with low risk of deaths in patients not receiving mechanical ventilation [6]. Monoclonal antibody therapy is currently evaluated at the clinical level and seems more efficient [7]. Both plasma and monoclonal antibodies seem to be efficient in decreasing deaths and hospitalizations when administered early in the disease course.

SARS-CoV-2 is more infectious than other coronaviruses and is deadly affecting middle aged and older people, resulting in a pandemic [8]. Many studies are underway to determine the number of people infected, to monitor the trend of infection in the adult general population, to determine the sociodemographic risk factors, and to delineate the geographical spread of the infection (seroprevalence) [9,10].

Serological antibody testing is critical for determining seroprevalence in a given population, defining previous exposure, and identifying highly reactive human donors for the generation of therapeutic convalescent serum [11,12]. Comparative phylogenomic analysis revealed several substitutions in the spike receptor-binding domain (S-RBD) required for adaptation to novel hosts [13,14]. The protein sequence alignment of the nucleocapsid protein (NP) provides evidence of sequence homology for some human coronaviruses [15]. Sensitive and specific identification of coronavirus SARS-CoV-2 antibody titers may, in the future, also support the screening of health care workers to identify those who are already immune and can be deployed to care for infected patients, thus minimizing the risk of viral spread to their colleagues and other patients. Overall, the validated assays described can be instrumental for the detection of SARS-CoV-2-specific antibodies for diagnostic, sero-epidemiologic, and vaccine evaluation studies.

The gold standard test for antiviral antibodies is the virus neutralization test [16], but in the case of SARS-CoV-2, high-containment laboratories are required. Pseudo-typed virus neutralization assays can be performed at low containment levels and read out with suitable reporters, such as luciferase, meaning that they are scalable [17]. Immunofluorescence (IF) can also use virus-infected cells to detect the presence of antibodies in the sample through their reaction with viral antigens expressed in the fixed cells without assessing the functionality of the antibodies.

Alternatively, ELISA and lateral flow assays (LFAs), which do not measure the function of the antibody, detect the binding of an antibody to a given antigen. The antigen is usually a recombinant protein, such as the whole spike protein, although some tests use a spike subdomain (S1) or a receptor-binding domain (RBD). Furthermore, antibody-mediated immunity against SARS-CoV-2 may have an effect on other endemic human coronaviruses that may arise in the future.

In the current study, we report the SARS-CoV-2 prevalence in regional civil protection (CP), police, healthcare personnel, and volunteers working in the COVID pandemic phase during the first wave and thanks to the B-LiFe mobile laboratory convenience. The objective of this study was primarily to assess the subjects’ seroconversion and secondarily to detect asymptomatic individuals within this cohort in the light of age, sex, and residence. The data reported here suggest that seroprevalence studies based on serological testing for S-RBD protein or SARS-CoV-2 NP antibodies are not sufficient for diagnosis but might help in determining the duration of seroconversion and in screening the population to be vaccinated.

## 2. Material and Methods

The concept of mass screening of citizens by specifically deployed mobile laboratories requires an efficient interconnectivity between the laboratory’s patient database and the host nation (known as the Laboratory Information Management system (LIMS)) and a connection with the national eHealth platform of another country. The algorithm for a smooth screening progress is described later.

Before starting the testing process, we administered a questionnaire (Table 1). All participants provided informed consent under the Italian Institutional Review Board(IRB)-approved protocol. A total of 6013 individuals (from first-responders) with ages ranging from 19 to 80 years underwent serological testing. The data were collected and analyzed using Cohen’s kappa statistical test (Supplementary data).

### Serological Test

The QuickZen^®^ COVID-19 IgM/IgG kit (ZenTech, Angleur, Belgium) is an immune colloidal gold test kit used for the qualitative detection of IgM and IgG against SARS-CoV-2 S-RBD in human whole blood or serum with a clinical sensitivity of 97% for IgM and 100% for IgG and clinical specificity reaching 99.1% for IgM and 97% for IgG (as claimed by the suppliers). It was used to screen a cohort population in Piedmont, Italy. Individuals found to be positive for SARS-CoV-2 S-RBD IgM, IgG, or both antibodies on whole blood finger-prick test were then swabbed for molecular testing and quantitative polymerase chain reaction (QPCR). The finger-prick blood samples were carefully dropped onto the sample wells of the strips. Given the fact that the SARS-CoV-2 virus is rarely detected in the blood of recovered donors or donors with mild/moderate symptoms, the blood drop presented a minimal biohazard risk [18,19]. The individuals with symptoms and positive results on serum antibody tests using the ZenTech kit [20] and validated by reverse transcription polymerase chain reaction (RT-PCR) [21] were deemed infected. Lateral flow tests were performed according to the manufacturer’s instructions and the results were read and interpreted within 10 min. Ten microliters of serum or whole blood sample was added to the sample port, followed by two drops of dilution buffer. The reagent binding pad was coated with colloidal gold-labeled recombination antigen, and rabbit IgG antibodies served as controls. The results were read visually after approximately 10 min.

NP, IgM, and IgG antibodies were detected by an immunochromatographic assay (SciMed, Brussels, Belgium) using a detection strip that included a sample pad, glass fiber (coated with fluorescent microspheres-antibody conjugates), and nitrocellulose membrane, absorbent paper, and PVC board, where the detection area T was coated with SARS-CoV-2 NP, and the quality control area was coated with sheep anti-mouse IgG. After the sample was added, the strip was placed aside for 15 min; next, the test strip was placed in a time-resolved IF analyzer to read the data. The IF analyzer measures and analyzes the optical signal to quantify the concentration of serum to be tested. The test result was the relative content of SARS-CoV-2 IgM or IgG antibodies. If the test result was less than 1 U/L, it was deemed negative; if the test result was more than or equal to 1 U/L, it was deemed positive.

Respiratory samples (Flocqswabs, Copan, Italy), serum, and plasma were collected from patients who were positive for IgM, IgG, or both. Total nucleic acids (DNA/RNA) were extracted from 200 µL of UTM and specific RT-PCR targeting RNA-dependent RNA polymerase, and E genes were used to detect the presence of SARS-CoV-2 according to the protocols [21].

## 3. Results and Discussion

We report the results of a study performed in Piedmont, Italy (Figure 1a) involving regional civil protection (CP), police, healthcare personnel, and volunteers working in the COVID pandemic phase. The population of this region was estimated according to the last official reports: 4,341,375 individuals with equal sex distribution and a large proportion of individuals (>60%) aged between 18 and 64 years (Figure 1b). A cohort of individuals with a similar sex and age distribution (Figure 1c) was tested using an algorithm (Figure 2), where individuals who tested positive for either IgM or IgG on the prick test were subjected to serum testing and molecular testing using nasopharyngeal swabs (RT-PCR) for COVID-19.

In the prick test, a total of 294 individuals from 6033 (4.8%) were positive for IgM and/or IgG (244 individuals (4%) for IgM, 126 individuals for IgG (2%), and 76 individuals for IgM and IgG (1.2%)). Furthermore, the seroconverted population was tested for viral particle load using RT-QPCR. No individual was positive on molecular testing, which is in line with the report of Cassaniti et al. [22] who reported that the IgM/IgG Rapid Lateral Flow immune assay (LFIA) is not recommended for triage of individuals suspected to have COVID-19 [23]. A potential explanation is the low viral particle titers or delayed humoral response. However, this may also highlight that viral particle load detection is not possible if seroconversion has already occurred.

The serum of seroconverted individuals (N = 294) was further analyzed for comparison between whole blood finger-prick and serum test results using a commercial test kit S-RBD distributed by QuickZen^®^, ZenTech). Regarding the IgG antibody detection, there was a strong inter-method agreement (Cohen’s kappa = 0.77) (Figure 3A) between the finger-prick and serum test results. In contrast, no agreement was noticed on IgM analysis (Cohen’s kappa = −0.11) (Figure 3B). One explanation is that seropositivity is highly influenced by the duration from symptom onset to seroconversion, which affects the sensitivity and specificity of strip assays based on LFIA [24].

Moreover, in serum state, the test results comparison for antibodies to SARS-CoV-2 NP obtained using a commercial test kit distributed by SciMed^®^ and produced by DIAGREAT^®^ [25], and S-RBD (QuickZen^®^, kit produced and distributed by ZenTech) showed a poor agreement in terms of IgM levels (Cohen’s kappa = −0.653) (Figure 3C) but not at the IgG levels where the agreement was substantial (Cohen’s kappa = 0.61) (Figure 3D). Ultimately, the data underlines that seroconversion in case of SARS-CoV-2 infection is neatly detected either by using blood finger-prick or serum for S-RBD QuickZen, Zentech kit. Consequently, the testing becomes a valuable tool to determine whether a successful vaccination program can lead to long-lasting immunization.

Many studies underline that the infection of the epithelial cells by SARS-CoV-2 is mediated by the interaction of SARS-CoV-2 S-RBD and angiotensin-I converting enzyme 2 (ACE-2) receptor [26,27,28]. Moreover, there is consensus that the antibodies directed against SARS-CoV-2 S-RBD are neutralizing antibodies [29,30,31,32,33]. Therefore, our data suggest that the detection of SARS-CoV-2 S-RBD antibodies (neutralizing antibodies) coincides with the presence of SARS-CoV-2 NP, which is in disagreement with the report of the researchers at Texas University (USA) [34]—their data suggested that the detection of N-protein-binding antibodies does not correlate with the presence of neutralizing antibodies. Therefore, they recommended caution against the extensive use of N-protein-based serology tests to determine potential COVID-19 immunity [34], as this may reflect exposure to SARS-CoV-2, but not protection against reinfection.

Until the present, no study has shown the protective effect of NP antibodies against reinfection. Still, many publications speculate that antibodies to the SARS-CoV-2 S-RBD neutralize the virus infection and are the best protectors against any potential reinfection with the same pathogen [28,35].

Most seroprevalence studies provide information only about previous exposure, rather than immunity, as no neutralizing antibodies are measured. In addition, the correlation between protection against SARS-CoV-2 infection and the titers of neutralizing antibodies that would protect recovered patients from secondary infection has not been formally defined; it is not clear if non-neutralizing antibodies (SARS-CoV-2 NP antibodies) would also protect against reinfection.

In comparison with common cold coronaviruses, immunity after COVID-19 seems to be incomplete and short-lived [36,37]. We also noticed that, as described by Long et al., asymptomatic SARS-CoV-2 patients show low antibody titers that wane quickly [38]. These patients are protected in one way or another by different immune functions, for instance, cellular immunity, that are not detected on serological assays, and this might alter the true exposure rate.

The neutralizing assay is the gold standard for serological diagnosis. In the case of COVID-19, recent exposure of the participants to common coronaviruses may have boosted the levels of SARS-CoV-2 neutralizing antibodies. However, cross-reactivity has been documented in other cases, such as zika and dengue viruses [39].

First responders with heavy COVID-19 exposure showed similar antibody prevalence to those with limited or no exposure. The guidelines on the use of personal protective equipment seem effective for preventing COVID-19 infection among healthcare workers.

In all, if the SARS-CoV-2 S-RBD antibodies could be detected as neutralizing antibodies on LFIA, which is very easy to distribute and use, it is pivotal to determine whether seroprevalence has downstream consequences for public health measures in estimating the case fatality and immunity.

Given these findings, the geographical variability of the incidence rates and dynamics of weekly seroprevalence rates during the early phase of the pandemic, these studies are only snapshots in time and space that reflect the circumstances of the period in which they were performed. We are caught in the wave of an unprecedented global health crisis, and such prevalence rates are of utmost importance for authorities to estimate exposure rates, especially in areas with little testing capacity; in case of vaccination drives, it will provide an idea about the extent and duration of vaccine-induced immunity.

SARS-CoV-2 exposure and infection may continue even if not on a pandemic scale unless an effective pharmacotherapy or vaccine is developed and administered. Lessons have been learned for future pandemics, and we need to be armed with a fundamental understanding of the interaction between host immunity and viruses for minimizing harm and optimizing favorable outcomes.

## Figures and Tables

**Figure 1 ijerph-18-03372-f001:**
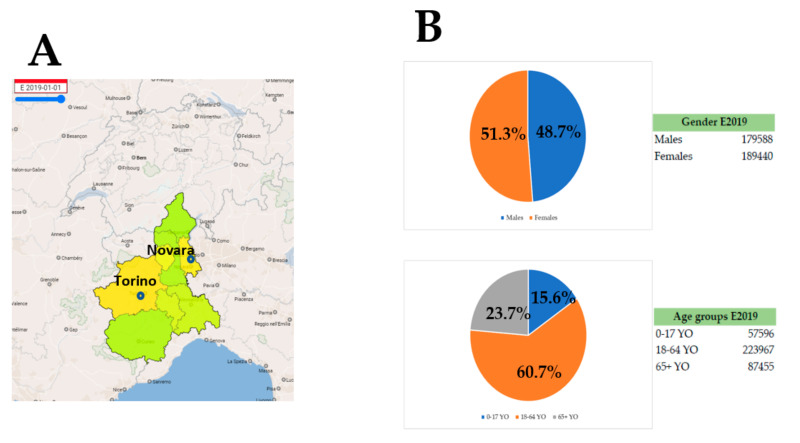
**(A**), Major cities of Piedmont, Italy involved in the current study (Torino and Novara) where colors define the population density/km^2^ (green: Low density and Yellow: high density). (**B**), Population structure of Piedmont according to Istituto Nazionale di Statistica Italia. (**C**), Age and sex distribution of the tested population.

**Figure 2 ijerph-18-03372-f002:**
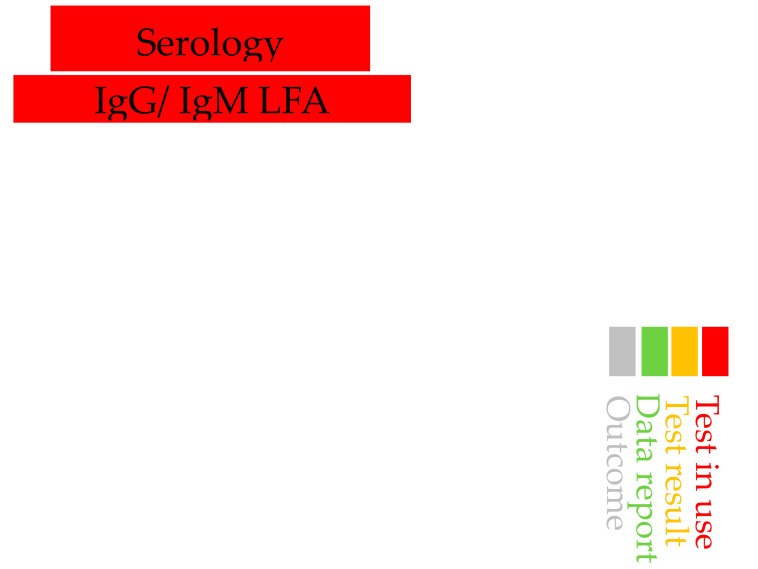
Testing algorithm schema applied for screening the population involved the current study.

**Figure 3 ijerph-18-03372-f003:**
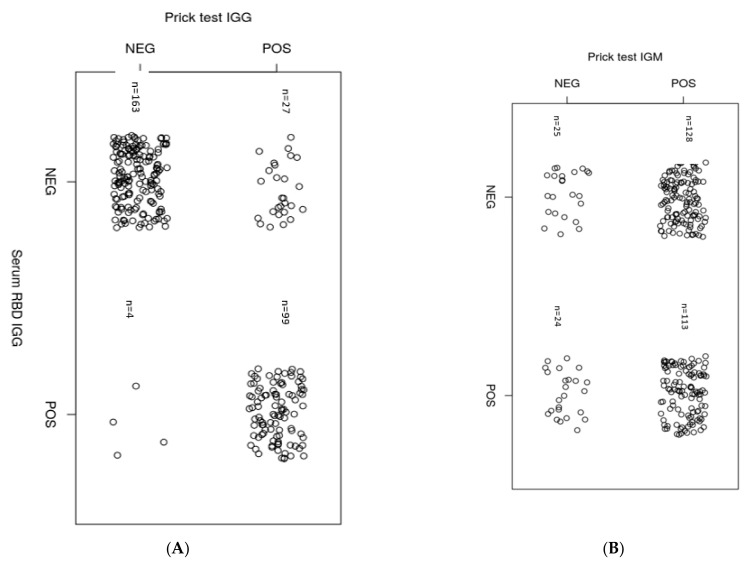
(**A**,**B**), Comparison of IgG and IgM levels on prick test versus serum against Spike S-RBD using serological tests. (**C**,**D**), Comparison of IgM and IgG levels against the nucleocapsid protein (NP) and spike receptor-binding domain (S-RBD) proteins in serum. To determine Cohen’s kappa coefficient (k) a statistical test was used.

**Table 1 ijerph-18-03372-t001:** Example of the questionnaire administered before testing.

Identity	National Registration Number
Date of first symptoms	Day/Month/Year
Date of last symptoms	Day/Month/Year
Cough	Yes/No
Anosmia	Yes/No
Dysgeusia	Yes/No
Headache	Yes/No
Muscle pain	Yes/No
Temperature	Yes/No
Respiratory distress	Yes/No

## Data Availability

Not publicly available.

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
