# Peer review of "Diagnostic Value of IgM and IgG Detection in COVID-19 Diagnosis by the Mobile Laboratory B-LiFE: A Massive Testing Strategy in the Piedmont Region"

_ijerph, 2021, doi:10.3390/ijerph18073372_

Round 1
Reviewer 1 Report
It is a nice work that was written in a short clean way. Maybe I was not reading carefully enough but I can not see the date of data used/showed in that work. Maybe it would be better to give it. At least it could be interesting that they were from the first wave of infection when there was only one kind of the virus. Or later when there was already two or more different mutants.
Figure 1 a: The mentioned big cities are shown is gray. Roughly speaking all the other cities are clearly visible except for the big cities wanted to be shown. Please put the name of Torino and Navara on the map in a more visible form. Please remove the thin black line from the right side of the picture (make a better cropping).
Figure 1 b: Is it really important to know how the population was changed from 1991 till 2020-01-01? Maybe the last (or the last two) column is important the population closest to the time of pandemy.
Figure 2: The figure is possibly a cropped part of a bigger one. The starting letter "F" is not fully visible. Also the side lines of the letters looks like stairs: we can see the pixels of the original letters.
At least the "F" in the title should be corrected. I think use of any graph making software you can make a new figure with much better real resolution. This form the Figure 2 does not look very serious.
Figure 3: The horizontal and vertical axis should have titles. The horizontal axis showing the age groups should be a solid (continous line) like the vertical axis showing the number of individuals in the given age range.
Figure4: The spell checker was on and marked the "badly spelled" tiles in the colored rectangles for example "Antigen"
So please remove the red underlines from the inserted texts.
Figure 6: The figure caption should be together (below) the figure. If you shrink a bit the graps of the page (Figure 5 and Figure 6) there will me enough space for the Figure 6 caption.
line 159-160:
The dot (.) should be at the end of the line 159.
Author Response
Reviewer 1:
It is a nice work that was written in a short clean way. Maybe I was not reading carefully enough but I cannot see the date of data used/showed in that work. Maybe it would be better to give it. At least it could be interesting that they were from the first wave of infection when there was only one kind of the virus. Or later when there were already two or more different mutants.
We thank the reviewer #1 for the appreciation of our study and for the comments that helped us improving our manuscript. In fact, the study was initiated and conducted during the first wave (early 2020) when no variant was reported.
Figure 1 a: The mentioned big cities are shown is gray. Roughly speaking all the other cities are clearly visible except for the big cities wanted to be shown. Please put the name of Torino and Navara on the map in a more visible form. Please remove the thin black line from the right side of the picture (make a better cropping).
The figure 1a has been revised accordingly
Figure 1 b: Is it really important to know how the population was changed from 1991 till 2020-01-01? Maybe the last (or the last two) column is important the population closest to the time of pandemy.
The figure 1b was omitted but related information was maintained in the main text
Figure 2: The figure is possibly a cropped part of a bigger one. The starting letter "F" is not fully visible. Also the side lines of the letters looks like stairs: we can see the pixels of the original letters.
At least the "F" in the title should be corrected. I think use of any graph making software you can make a new figure with much better real resolution. This form the Figure 2 does not look very serious.
We thank the reviewer for his advice. The figure 2 has been revised accordingly
Figure 3: The horizontal and vertical axis should have titles. The horizontal axis showing the age groups should be a solid (continuous line) like the vertical axis showing the number of individuals in the given age range.
The figure 3 (current Figure 1C) has been revised accordingly
Figure 4: The spell checker was on and marked the "badly spelled" tiles in the colored rectangles for example "Antigen" So please remove the red underlines from the inserted texts.
The figure 4 (current figure 2) has been revised accordingly
Figure 6: The figure caption should be together (below) the figure. If you shrink a bit the graps of the page (Figure 5 and Figure 6) there will me enough space for the Figure 6 caption.
We thank the reviewer for his advice. The figure has been revised accordingly
line 159-160: The dot (.) should be at the end of the line 159.
This has been corrected
Reviewer 2 Report
Omar et al are describing the diagnostic value of IgM and IgG detection in COVID-19 diagnosis in Piedmont in Italy.
Major points:
Introduction:
The introduction misses why this study is important, the objectives as well as a summary of what was done in this work. The objectives are only mentioned in the abstract but not in the introduction. Line 53 to 65 - Citations are missing. Please cite relevant literature to underline your statements.
Figures:
Overall, the figures look like screenshots and are in bad quality. The figures are missing a figure legend and explanation of abbreviations.
Figure 1 - This figure has not been compiled to one – A and B are presented separately and it looks still preliminary.
Figure 1a - The figure shows areas in different shades of green. What does it mean, please elaborate in the legends or manuscript. It is difficult to see the two cities on this map. Why is it important to show the location of both cities? Please change it and elaborate in the manuscript.
Figure 1b - Elaborate why is it important to see the population of all other Italian regions. I suggest, the information regarding the population in Piedmont could be only given in the text. The figure looks like a screenshot from a website. Moreover, this is not a figure but it should be presented as a table. The square which is marking Piedmont seems to be shifted to the left.
Figure 2 - The figure has been cut off on the left side and looks again like a screenshot from a website.
Figure 3 - The y-axis is too short for the presented data. The colors green and green are also not optimal chosen. I suggest a stronger contrast for color blind people. The legends are not explaining what is x and what is y-axis.
Figure 1, 2 and 3 are not in proportion for the information they give in the manuscript. It would be more than sufficient to combine them into one figure.
Figure 4 - The quality of the figure is very bad. In addition, there are some words underlined in red waves. What does it mean? Or is this the language correction? Please if you use abbreviations of words, explain them in the legends.
Figure 5,6,7 and 8 could be A, B, C and D. Figure legends are missing here as well. In addition, each quadrant should have the actual numbers written in it (e.g. n=4) or the percentage to give more accurate information to the reader.
References: Numbering is missing.
Minor points:
Line 11 - Please state the year so it will be clear for readers in the future.
Line 31 - I don't agree that SARS-CoV-2 is resistant to therapy. Antibody or convalescent plasma therapy has been successful applied as treatment. Otherwise, please rephrase or write why it would be considered resistant to therapy?
Line 36 - Please rephrase that sentence.
Table 1 - D/M/Y and Y/N is supposed to be the date and Yes/No? Please mention it in the table legend.
Line 84 - Quantitative polymerase chain reaction is written in bold
Line 172 to 175 – “The data is in disagreement with a report from the Texas University.” Please cite and elaborate.
Line 190 - Long et al 2020 is not in the reference list.
Figures - If information was retrieved from a website, please cite the exact website and the date when the data was retrieved from it.
Overall the submitted manuscript looks very preliminary and not suitable for publication in the present form.
Author Response
Reviewer 2:
Omar et al are describing the diagnostic value of IgM and IgG detection in COVID-19 diagnosis in Piedmont in Italy.
We thank the reviewer #2 for the appreciation of our study and for the relevant comments that helped us improving our manuscript.
Major points:
Introduction:
The introduction misses why this study is important, the objectives as well as a summary of what was done in this work. The objectives are only mentioned in the abstract but not in the introduction. Line 53 to 65 - Citations are missing. Please cite relevant literature to underline your statements.
We thank the reviewer for his comments.
The rationale and objectives have been added in the last part of the introduction section.
References have also been inserted in the corresponding section as requested.
Figures:
Overall, the figures look like screenshots and are in bad quality. The figures are missing a figure legend and explanation of abbreviations.
All the figures have been revised and relooked. The legends of the figures have also been rewritten
Figure 1 - This figure has not been compiled to one – A and B are presented separately and it looks still preliminary.
This has been corrected
Figure 1a - The figure shows areas in different shades of green. What does it mean, please elaborate in the legends or manuscript. It is difficult to see the two cities on this map. Why is it important to show the location of both cities? Please change it and elaborate in the manuscript.
The figures have been revised and relooked. The legends of the figures have also been rewritten
Figure 1b - Elaborate why is it important to see the population of all other Italian regions. I suggest, the information regarding the population in Piedmont could be only given in the text. The figure looks like a screenshot from a website. Moreover, this is not a figure but it should be presented as a table. The square which is marking Piedmont seems to be shifted to the left.
We agree. The table was omitted and related information was kept in the main text
Figure 2 - The figure has been cut off on the left side and looks again like a screenshot from a website.
All the figures have been revised and relooked. The legends of the figures have also been rewritten
Figure 3 - The y-axis is too short for the presented data. The colors green and green are also not optimal chosen. I suggest a stronger contrast for color blind people. The legends are not explaining what is x and what is y-axis.
All the figures have been revised and relooked. The legends of the figures have also been rewritten
Figure 1, 2 and 3 are not in proportion for the information they give in the manuscript. It would be more than sufficient to combine them into one figure.
We thank the reviewer for his advice. We did merge figures 1 and 2.
Figure 4 - The quality of the figure is very bad. In addition, there are some words underlined in red waves. What does it mean? Or is this the language correction? Please if you use abbreviations of words, explain them in the legends.
All the figures have been revised and relooked. The legends of the figures have also been rewritten
Figure 5,6,7 and 8 could be A, B, C and D. Figure legends are missing here as well. In addition, each quadrant should have the actual numbers written in it (e.g. n=4) or the percentage to give more accurate information to the reader.
All the figures have been revised and relooked. The legends of the figures have also been rewritten
References: Numbering is missing.
The list of references has been revised
Minor points:
Line 11 - Please state the year so it will be clear for readers in the future.
This has been corrected
Line 31 - I don't agree that SARS-CoV-2 is resistant to therapy. Antibody or convalescent plasma therapy has been successful applied as treatment. Otherwise, please rephrase or write why it would be considered resistant to therapy?
The sentence has been rephrased for more clarity and accuracy (Page 01, Line 34)
Line 36 - Please rephrase that sentence.
The sentence has been rephrased for more clarity and accuracy
Table 1 - D/M/Y and Y/N is supposed to be the date and Yes/No? Please mention it in the table legend.
The table has been revised
Line 84 - Quantitative polymerase chain reaction is written in bold
This has been corrected
Line 172 to 175 – “The data is in disagreement with a report from the Texas University.” Please cite and elaborate.
The related reference has now been inserted and their data more discussed
Line 190 - Long et al 2020 is not in the reference list.
This has been inserted
Figures - If information was retrieved from a website, please cite the exact website and the date when the data was retrieved from it.
All the figures have been revised and relooked. The legends of the figures have also been rewritten

Reviewer 3 Report
Omar et al present a prospective study evaluating the use of a rapid COVID-19 finger-prick test in the Piedmont region of Italy, an area that was severely affected by the first wave of infection. They compare results to serum antibody testing. The manuscript is of interest, however the figures are poorly collated and need to include more of the results and less of the background information. As well, the paper would be enhanced by more clinical comparisons between the positive and negative group, however if this is not possible due to data availability, I do not think it is a reason to reject the paper. It could be significantly shortened in that case then as the results are really less than half a page.
Minor issues:
- Figure 1a is reasonable to include but Figure 1b could be summarized as a single line in the methods or introduction and is not necessary.
- Figure 2 has part of the edge cut off from cropping. Please adjust if kept, however I also think this could be removed from the manuscript
- Figure 5-8 should all be condensed into one figure.
- Figure 4 can be kept but they present a screen shot that has the red underline from a Microsoft Office software showing a spellcheck error in multiple locations.
- It is unclear when the authors move from results to discussion as there is no header. Please adjust.
- I would like to see more statistics looking at what patient characteristics are present based on those who are positive vs negative. This is a very interesting cohort, however the authors present almost no clinical correlation. If this is unavailable due to data availability it would not be a reason to reject the paper but is unfortunate.
- Please include statistical tests inside the figures that have between group comparisons for assays.
Author Response
Reviewer 3:
Omar et al present a prospective study evaluating the use of a rapid COVID-19 finger-prick test in the Piedmont region of Italy, an area that was severely affected by the first wave of infection. They compare results to serum antibody testing. The manuscript is of interest, however the figures are poorly collated and need to include more of the results and less of the background information. As well, the paper would be enhanced by more clinical comparisons between the positive and negative group, however if this is not possible due to data availability, I do not think it is a reason to reject the paper. It could be significantly shortened in that case then as the results are really less than half a page.
We thank the reviewer #3 for the appreciation of our study and for the relevant comments that helped us improving our manuscript.
Minor issues:
- Figure 1a is reasonable to include but Figure 1b could be summarized as a single line in the methods or introduction and is not necessary.
This has been changed
- Figure 2 has part of the edge cut off from cropping. Please adjust if kept, however I also think this could be removed from the manuscript
This has been corrected
- Figure 5-8 should all be condensed into one figure.
This has been changed
- Figure 4 can be kept but they present a screen shot that has the red underline from a Microsoft Office software showing a spellcheck error in multiple locations.
This has been corrected
- It is unclear when the authors move from results to discussion as there is no header. Please adjust.
- Our paper is describing a case study in which a diagnosis method was applied. Hence, we believe it will make more sense, coherence and interest if the results of the methods will be presented and directly interpreted. I would like to see more statistics looking at what patient characteristics are present based on those who are positive vs negative. This is a very interesting cohort, however the authors present almost no clinical correlation. If this is unavailable due to data availability it would not be a reason to reject the paper but is unfortunate.
We thank the reviewer#3 for this relevant comment. Unfortunately, the data were not available at the time of the study due to the emergency situation.
- Please include statistical tests inside the figures that have between group comparisons for assays.
We thank the reviewer#3 for this relevant comment. All remarks have been considered and adjusted accordingly While statistical tests were provided in supplementary data.

Round 2
Reviewer 2 Report
I thank the authors for accepting my suggestions for their manuscript.
Minor points:
Line 13: There is a underscore before - "Herein,..."
The text has both version: SARS-CoV-2 and SARS-Cov-2. Please unify it.
Line 392: There is a double bracket
Line 404: The size of the RBD is mentioned. Why is that mentioned here?
Line 405: The quotes are separate by a comma.
Major points:
Line 33: There are different kind of therapeutics besides steroids, which have been shown to work well, such as monoclonal antibodies https://www.nature.com/articles/d41586-021-00650-7 and convalescent plasma therapy https://www.nejm.org/doi/full/10.1056/NEJMoa2031893. Therefore, the statement in the introduction would be misleading. Monoclonal antibody therapy and convalescent plasma therapy have received an emergency use approval by the FDA in the US.
Figure 1A, B and C are not combined into one figure.
Figure 1A - Could the authors please explain what the green areas on the map mean? This is not explained in the text.
Figure 2 is cut-off on the side.
Author Response
Reviewer-2
Minor points:
Line 13: There is a underscore before - "Herein,..."
This has been corrected
The text has both version: SARS-CoV-2 and SARS-Cov-2. Please unify it.
This has been corrected
Line 392: There is a double bracket
This has been corrected
Line 404: The size of the RBD is mentioned. Why is that mentioned here?
We agree the information is not useful in that context. This has been omitted.
Line 405: The quotes are separate by a comma.
This has been corrected
Major points:
Line 33: There are different kind of therapeutics besides steroids, which have been shown to work well, such as monoclonal antibodies https://www.nature.com/articles/d41586-021-00650-7 and convalescent plasma therapy https://www.nejm.org/doi/full/10.1056/NEJMoa2031893. Therefore, the statement in the introduction would be misleading. Monoclonal antibody therapy and convalescent plasma therapy have received an emergency use approval by the FDA in the US.
We fully agree with the reviewer #2 although this information was very recent, reason why it was not included. We have now cited these data in the introduction (page 1, lines 36-44).
Figure 1A, B and C are not combined into one figure.
This has been corrected
Figure 1A - Could the authors please explain what the green areas on the map mean? This is not explained in the text.
The green area in the map means the low density/km2. For more details https://www.citypopulation.de/en/italy/cities/piemonte/
Figure 2 is cut-off on the side.
This has been corrected

Reviewer 3 Report
Thank you for the revised manuscript. Unfortunately, many of the major comments were not possible to address due to data availability, however I think the manuscript still warrants publication.
Minor Comments:
1) At the end of the first paragraph of methods the authors describe an algorithm that was "outlined further." Can they please describe where it is outlined further.
2) When I requested Figure 1 be revised, I meant so that it would fit on one page, not that it should be revised in a way so that just the labels are changed. Please create a multi-pane figure that would fit on one 8.5 x 11 page (this comment can be ignored at the discretion of the editor).
Author Response
Reviewer3
Minor Comments:
1) At the end of the first paragraph of methods the authors describe an algorithm that was "outlined further." Can they please describe where it is outlined further.
We apologize for this mistake. The algorithm is described in figure 2. This statement has been corrected
2) When I requested Figure 1 be revised, I meant so that it would fit on one page, not that it should be revised in a way so that just the labels are changed. Please create a multi-pane figure that would fit on one 8.5 x 11 page (this comment can be ignored at the discretion of the editor).
We thank the reviewer #3 for his comment. We have revised and reorganized this figure 1 to make our statements more clear and straightforward
